# BD-Func: a streamlined algorithm for predicting activation and inhibition of pathways

Charles D. Warden[1], Noriko Kanaya[2], Shiuan Chen[2] and Yate-Ching Yuan[1]

[1] Bioinformatics Core, Department of Molecular Medicine, Duarte, CA, United States
[2] Department of Cancer Biology, City of Hope National Medical Center, Duarte, CA, United States

## ABSTRACT

BD-Func (BiDirectional FUNCtional enrichment) is an algorithm that calculates functional enrichment by comparing lists of pre-defined genes that are known to be activated versus inhibited in a pathway or by a regulatory molecule. This paper shows that BD-Func can correctly predict cell line alternations and patient characteristics with accuracy comparable to popular algorithms, with a significantly faster run-time. BD-Func can compare scores for individual samples across multiple groups as well as provide predictive statistics and receiver operating characteristic (ROC) plots to quantify the accuracy of the signature associated with a binary phenotypic variable. BD-Func facilitates collaboration and reproducibility by encouraging users to share novel molecular signatures in the BD-Func discussion group, which is where the novel progesterone receptor and LBH589 signatures from this paper can be found. The novel LBH589 signature presented in this paper also serves as a case study showing how a custom signature using cell line data can accurately predict activity *in vivo*. This software is available to download at https://sourceforge.net/projects/bdfunc/.

## INTRODUCTION

Systems-level analysis of the combined expression pattern of multiple genes can be more informative than the expression pattern of an individual gene, and there are a number of tools to calculate functional enrichment of differentially expressed genes (*Huang, Sherman & Lempicki, 2009*; *Naeem et al., 2012*; *Nam & Kim, 2008*). However, many functional annotations merely list membership in a pathway or ontology without explicitly modelling genes that should show activation or inhibition. For example, consider the KEGG canonical Wnt signalling pathway (Fig. 1) (*Kanehisa & Goto, 2000*). This gene list includes molecules that both activate and inhibit the pathway, resulting in different phenotypes (*Dellinger et al., 2012*; *Logan & Nusse, 2004*). However, many functional enrichment tools would expect all the members of the pathway to behave similarly (Fig. 1C), such that up-regulation of a mix of activators and inhibitors can receive a higher score than selective up-regulation of only activators within the pathway. For example, the most standard

Corresponding authors
Charles D. Warden,
cwarden@coh.org
Yate-Ching Yuan, yyuan@coh.org

**A.** Complete Activation    **B.** Complete Inhibition    **C.** Mixed Pattern

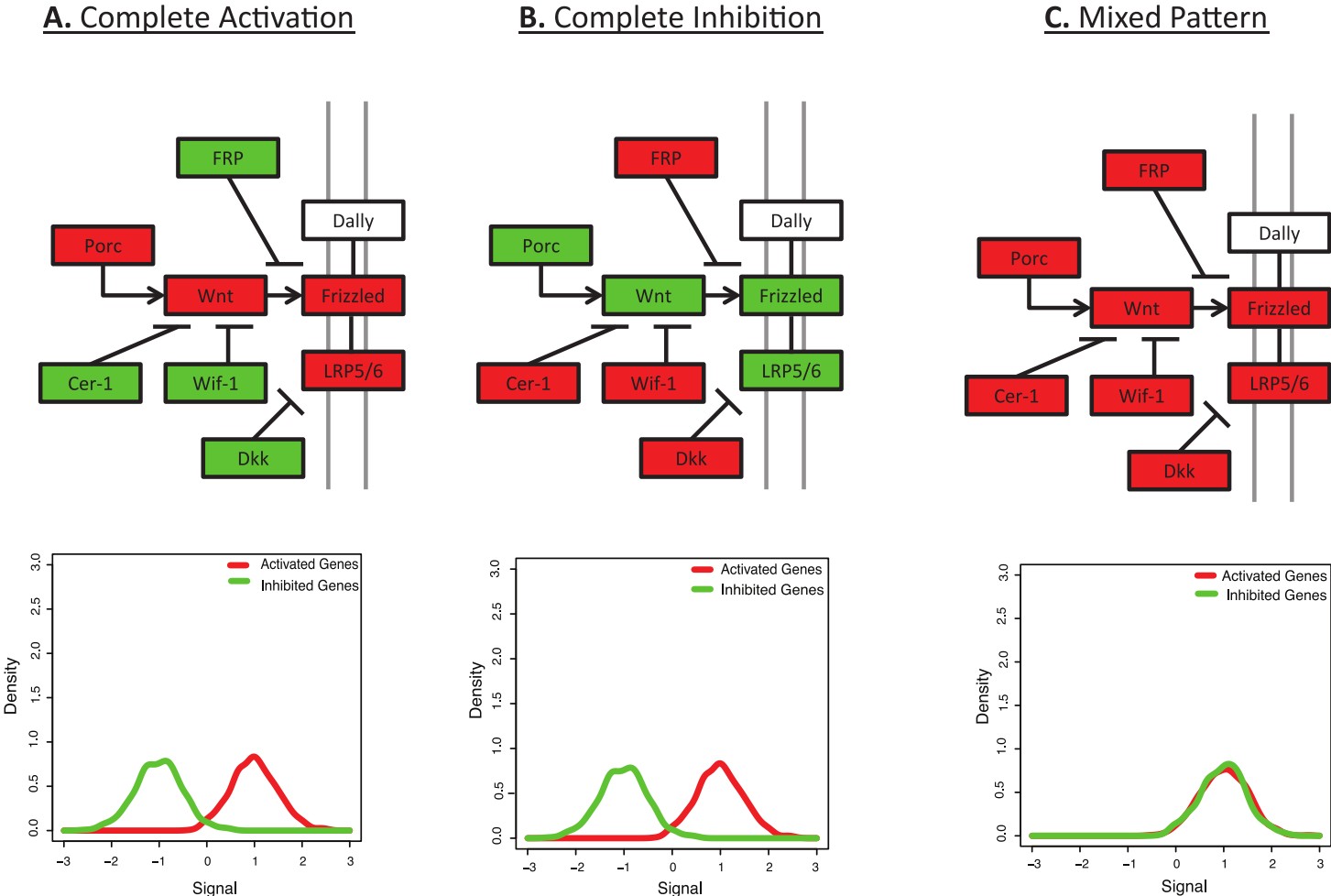

**Figure 1** **Many pathways are characterized by a mix of activation and inhibition.** This figure shows the initial signalling steps in the Wnt signalling pathway, as defined by KEGG (*Kanehisa & Goto, 2000*). Up-regulated genes are shown in red and down-regulated genes are shown in green. (A) Complete Activation: All genes activating the Wnt signalling pathway are up-regulated and all inhibitors are down-regulated. (B) Complete Inhibition: All genes activating the Wnt signalling pathway are down-regulated and all inhibitors are up-regulated. (C) Mixed Pattern: All genes in the figure are up-regulated. It is unclear what the downstream expression levels should be, but one may hypothesize a mixed result from Figs. 1A and 1B. However, most functional enrichment tools would predict this as the pattern with the strongest up-regulation. Underneath each diagram is the expected signal distribution that would be produced by BD-Func.

method for functional enrichment is to calculate over-representation of one gene list in another gene list, possibly using a Fisher's exact test or hypergeometric test; in the example described above, this sort of statistical test would ask if a list of differentially expressed genes shows a higher number of Wnt signalling genes than expected by chance. This sort of test cannot differentiate the behavior of those genes unless more detailed gene lists are defined (such as Wnt-inhibitors versus Wnt-agonists). This is a basic problem that BD-Func (BiDirectional FUNCtional enrichment) is designed to overcome.

Most functional enrichment tools either require upstream filtering of gene lists (FuncAssociate (*Berriz et al., 2009*), GATHER (*Chang & Nevins, 2006*), DAVID (*Huang, Sherman & Lempicki, 2008*), Connectivity Map (*Lamb et al., 2006*), WebGestalt

(*Zhang, Kirov & Snoddy, 2005*), GoMiner (*Zeeberg et al., 2003*), ErmineJ (*Lee et al., 2005*))
and/or a comparison of signal intensities between two groups (T-profiler (*Boorsma et
al., 2005*), GSVA (*Hanzelmann, Castelo & Guinney, 2013*), PAGE (*Kim & Volsky, 2005*),
GSEA (*Subramanian et al., 2005*), ErmineJ (*Lee et al., 2005*)). However, BD-Func compares
the relative expression levels between activated and inhibited genes, and we show that
BD-Func can successfully analyze either fold-change values between populations or raw
intensity/expression values (for both microarray and RNA-Seq data). Signalling Pathway
Impact Analysis (*Draghici et al., 2007*; *Tarca et al., 2009*) can be used to model activation
and inhibition within a graph, but that algorithm primarily focuses on network topology
(which is not always known); in contrast, BD-Func uses a simpler assumption of binning
genes into two categories (activation or inhibition). Additionally, the ability to analyze
absolute expression values in a single sample is a unique feature only present in a limited
number of functional genomic tools. For example, ASSESS (*Edelman et al., 2006*) can
theoretically predict functional enrichment in a single sample, but that score is not taken
into context amongst other samples: in other words, BD-Func uniquely uses a single
sample enrichment score as a classifier, which may be useful for personalized medicine
research or for the development of novel molecular signatures where the user may need
to quantify the utility of the signature as a classifier. Additionally, correlations between
a single sample and various samples within a database (such as SPIED (*Williams, 2012*))
can provide information about a single sample, but this requires having a database of
samples for comparison (so, this strategy will not work without the presence of an external
database). Finally, BD-Func users are encouraged to share their lists of activated and
inhibited genes in a simple file format. This allows easy application of models that may not
be in an existing database for molecular signatures.

This study tests the accuracy of BD-Func on datasets that were used to define gene sets
in MSigDB (Molecular Signatures DataBase, *Subramanian et al., 2005*), in comparison
to GSEA (Gene Set Enrichment Analysis, *Mootha et al., 2003*; *Subramanian et al., 2005*)
and IPA (Ingenuity Pathway Analysis, Ingenuity® Systems, www.ingenuity.com). IPA
was selected for comparison because the upstream regulator function utilizes a similar
principle as BD-Func (activation and inhibition is predicted by comparing the proportion
of activated or inhibited targets, based upon annotations in the proprietary IPA database).
GSEA was selected for comparison for two reasons: (1) GSEA was specifically designed to
analyze MSigDB signatures, thus serving as a good baseline for positive control datasets
and (2) MSigDB contains some signatures for both up- and down-regulated genes,
so it is useful to compare separate analysis of these signatures (using GSEA) versus a
direct comparison of up-regulated to down-regulated gene expression (using BD-Func).
Different models for TGF$\beta$ and progesterone receptor (PGR) activity are also tested for
robustness by application to other datasets. Finally, the utility of BD-Func to study custom
gene signatures is tested with a novel signature associated with progesterone receptor
status in breast cancer patients as well as a novel LBH589 signature that was defined using
previously published cell line data and is validated using novel *in vivo* data presented in this
study.

## MATERIAL AND METHODS

### BD-Func algorithm

The basic principle behind BD-Func is to treat activated and inhibited genes as replicate observations in two populations. BD-Func is agnostic towards the type of signal used for analysis. For example, this paper uses BD-Func to study both fold-change values between two populations as well as raw signal intensities for a single column of signal values. In the paper, the *t*-test statistic is used to compare the expression patterns of activated and inhibited genes. However, BD-Func also allows users to compare the activated and inhibited distributions using a Mann–Whitney U test or Kolmogorov–Smirnov (K–S) test. BD-Func users have the option to calculate False Discovery Rates (FDR) using the method of *Benjamini & Hochberg (1995)* or the Storey *q*-value (*Storey & Tibshirani, 2003*).

BD-Func comes with four enrichment files: c2, c5, and c6 from MSigDB (*Liberzon et al., 2011*) and a list of functions defined directly from the human Gene Ontology (GO) annotations (*Ashburner et al., 2000*). All of these lists were created by searching for functions with both "up" and "down" (or "positive regulation" and "negative regulation") gene lists. For the human GO file, a functional annotation needed to contain at least 10 positively regulated genes and 10 negatively regulated genes. We also encourage users to share their own custom models on the BD-Func discussion group: http://sourceforge.net/p/bdfunc/discussion/.

There are three different input files that can be analyzed using BD-Func (Fig. S1):

*1-D Input File*: If the user supplies expression values for a single column of data, a density plot is created for the signal for the activated and inhibited genes. In this case, the BD-Func algorithm works exactly as described above.

*2-D Input File*: If the input file contains multiple columns of data, BD-Func is first run separately for each sample (represented by a column in the data matrix), as described above. Next, box-plots are created for test-statistic scores for each group (labelled in the header of the input file; Fig. S2). Finally, an ANOVA *p*-value is provided to compare the test statistics between groups. Theoretically, test statistics could be used to make functional predictions for each sample in isolation. There are some examples in this paper where this strategy works OK. However, comparing test statistics across all samples within different groups is the only strategy that consistently works well for all the analysis presented in this paper.

*2-D Input File for Classifier*: If the input file contains multiple columns of data and two groups (called "positive" and "negative"), BD-Func will create a receiver operating characteristic (ROC) plot using the test-statistic as the classification score using the ROCR package (*Sing et al., 2005*). This is in addition to all the calculations and output files for a normal 2-D input file containing multiple columns for BD-Func to analyze (in this paper, this represents normalized signal intensity values).

### Sample acquisition and processing

Microarray datasets were downloaded from GEO (*Edgar, Domrachev & Lash, 2002*) or Array-Express (*Parkinson et al., 2009*). When raw CEL files were available, samples were

RMA normalized (*Irizarry et al., 2003*). Otherwise, processed intensity values were used for microarray analysis. Fold-Change values for all of the cell line datasets (TGF$\beta$ (*Padua et al., 2008*; *Qin et al., 2009*; *Renzoni et al., 2004*; *Sartor et al., 2010*; *Scandura et al., 2004*), mTOR (*Wei et al., 2006*), p53 (*Elkon et al., 2005*), and BRCA1 (*Furuta et al., 2006*)) and the MSigDB progesterone receptor dataset (*Claus et al., 2008*) were calculated using the method of the least-squares mean using Partek® Genomics Suite™ (*Partek Inc, 2012*).

All other clinical samples (*Anders et al., 2008*; *Bild et al., 2006*; *Chin et al., 2006*; *Finak et al., 2008*; *Huang et al., 2003*; *Ivshina et al., 2006*; *Sotiriou et al., 2003*; *The Cancer Genome Atlas Network, 2012*) were downloaded and analyzed for differential expression using BRAVO (http://bravo.coh.org/) (X Deng, C Warden, Z Liu, I Zhang, Y-C Yuan, unpublished data). The novel progesterone receptor gene signature presented in this paper was produced by identifying genes in the expO dataset (GEO Series GSE2109) with a |fold-change| > 2 and an False Discovery Rate (FDR) < 0.05 (where the FDR is calculated using the method of *Benjamini & Hochberg (1995)* to analyze the distribution of $t$-test $p$-values). This is how the samples in this particular paper were processed, but users are not required to use this particular set of tools for preparing BD-Func input files and/or creating gene lists for custom signatures.

Reads Per Kilobase per Million mapped reads (RPKM (*Mortazavi et al., 2008*)) values for RNA-Seq data was downloaded from the TCGA web portal (*The Cancer Genome Atlas Network, 2012*). RPKM values were transformed by addition of 0.1 (to avoid large fold-change values for low coverage reads) followed by a $\log_2$ transformation (to normalize the signal distribution).

## GSEA comparison

With the exception of the progesterone receptor signature (which utilized the CLAUS_PGR_POSITIVE_MENINGIOMA (*Claus et al., 2008*) signature from MSigDB-c2, version 3.1), oncogenic signatures were defined using the following gene lists from MSigDB-c6 (*Liberzon et al., 2011*, version 3.1) for GSEA analysis: TGFB_UP.V1 (*Padua et al., 2008*), MTOR_UP.N4.V1 (*Wei et al., 2006*), P53_DN.V2 (*Elkon et al., 2005*), and BRCA_DN.V1 (*Furuta et al., 2006*). GSEA (*Subramanian et al., 2005*, version 2.0) calculated $p$-values by permutation over phenotypes whenever possible (*Anders et al., 2008*; *Chin et al., 2006*; *Claus et al., 2008*; *Elkon et al., 2005*; *Finak et al., 2008*; *Huang et al., 2003*; *Padua et al., 2008*; *Sartor et al., 2010*; *Scandura et al., 2004*; *The Cancer Genome Atlas Network, 2012*; *Wei et al., 2006*), although there were a few datasets with less than 3 replicates for which gene sets had to be permuted instead of phenotypes (*Furuta et al., 2006*; *Qin et al., 2009*; *Renzoni et al., 2004*). For recovery of known perturbations of oncogenic regulators, GSEA results must either show a FWER $p$-value < 0.25 or a NOM $p$-value < 0.05, which are the default cut-offs.

## IPA comparison

Ingenuity Pathway Analysis (IPA; Ingenuity® Systems, www.ingenuity.com) contains an "Upstream Regulator" module that compares the enrichment of activated and inhibited

genes among up- and down-regulated genes. So, the underlying principle is similar to BD-Func except it utilizes Ingenuity's propriety database of regulatory interactions and uses a $z$-score to calculate statistical significance between activated and inhibited genes. Gene lists in IPA were filtered for those genes showing |fold-change| > 1.5 while the entire gene list is used to define background enrichment. For recovery of known perturbations of oncogenic regulators, the upstream regulator must be identified as "activated" or "inhibited" (|$z$-score| > 2), which are the default cut-offs.

### LBH589 signature

Activated and inhibited genes were defined using overlapping gene lists from 3 cell line treatments that have been previously published (*Kubo et al., 2013*). That same study showed that LBH589 treatment significantly decreased tumor volume in exemestane (EXE) resistant MCF-7aro xenografts in mice. This study analyzes novel microarray data from EXE-resistant tumors treated with (EXE + LBH589) or without (EXE only) treatment of LBH589. All animal research procedures were approved by the City of Hope Institutional Animal Care and Use Committee. This novel microarray data is available in GEO series GSE47346.

In order to be included in the novel BD-Func signature, genes must show differential expression in all 3 cell lines. Genes were defined as differentially expressed if they showed a |fold-change| > 1.5 and $p$-value < 0.05 , and the BD-Func signature genes had to meet these conditions for each of the 3 LBH589 cell line treatments (with consistent direction of fold-change). *P*-values were calculated via 1-way ANOVA with appropriate linear contrast was used to compare data sets using Partek® Genomics Suite™ (Partek, Inc., St. Louis, MO). Fold-change values were calculated based upon the least-squares mean, and data was normalized using robust multichip average (RMA) normalization (*Irizarry et al., 2003*).

## RESULTS

### BD-Func shows equal or greater performance to GSEA and IPA for functional enrichment

Given the relative ease by which samples can be classified as having positive or negative activity for an individual biomarker, the accuracy of BD-Func was first tested by applying several MSigDB oncogenic signatures (*Liberzon et al., 2011*) to the datasets from which the signatures were defined (*Claus et al., 2008*; *Elkon et al., 2005*; *Furuta et al., 2006*; *Padua et al., 2008*; *Wei et al., 2006*). BD-Func was able to detect the activation or inhibition of all of these oncogenic signatures (Table 1, Fig. 2). GSEA could detect all of the signatures except the *Claus et al. (2008)* progesterone receptor signature. Among these 5 test datasets, IPA could only detect the activity of 2 of these genes; however, this is not a completely fair comparison because we would expect some overfitting of the MSigDB signatures for the GSEA and BD-Func analysis. Nevertheless, the significance of this analysis is that BD-Func can accurately detect perturbation of all of these biomarkers on datasets where we know that these specific genes will be altered.

In order to test the performance of BD-Func, GSEA, and IPA on novel datasets (which were not used by MSigDB to define gene lists), we applied the 3 algorithms to four datasets

A.

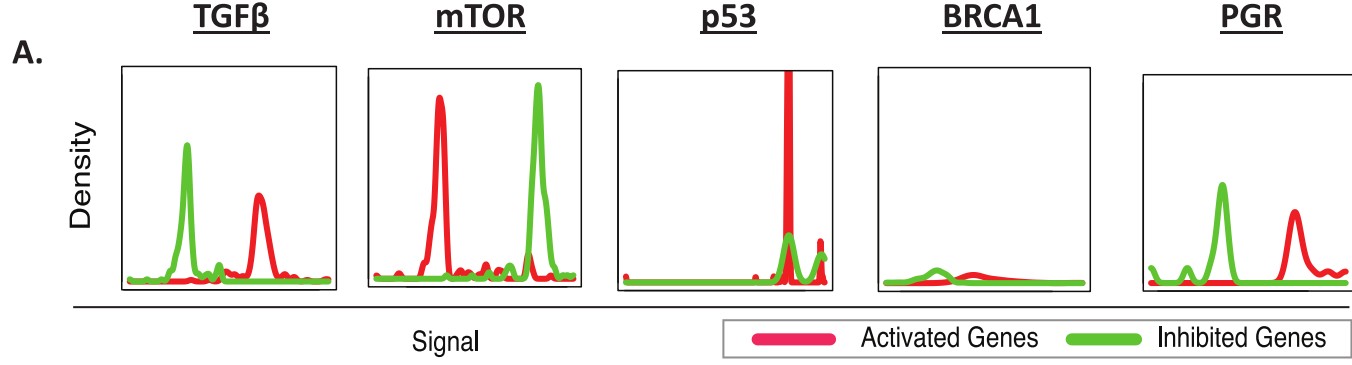

B.

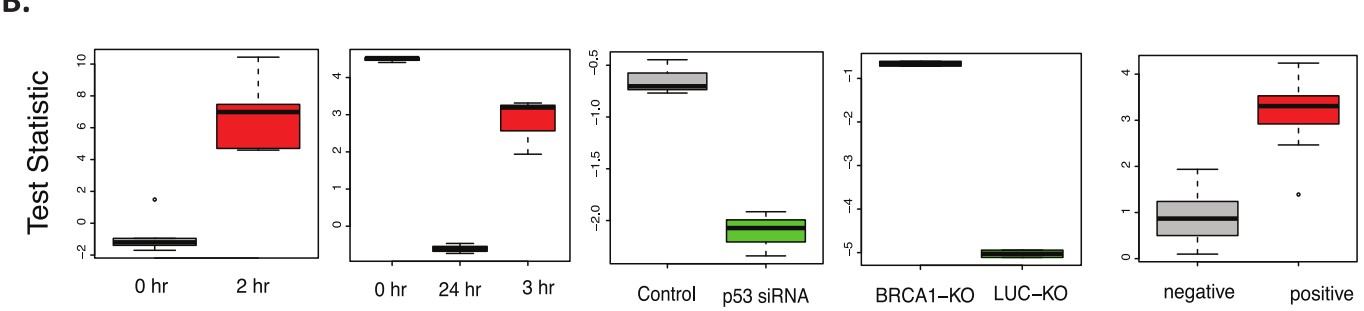

**Figure 2** **BD-Func recovers known MSigDB signatures.** This figure shows the output figures from BD-Func for the 5 MSigDB signatures tested on their original datasets. (A) Density plots for fold-change values for activated genes (colored red) and inhibited genes (colored green). These plots are used to illustrate BD-Func analysis on a single column of data (in this case, fold-change values between the positive and negative populations). (B) Box-plots of activation versus inhibition test statistics for all relevant samples in each of the 5 MSigDB datasets. Note that each box-plot describes the distribution of *test statistics* for each group – it does not represent the expression of an individual gene or a metagene. In each of these five examples, the test statistic shows very clear differences among the different groups. If the median *t*-test statistic is greater than 2, the box is colored red. If the median *t*-test statistic is less than −2, the box is colored green.

with TGF$\beta$ treatments (Table 2). All 3 algorithms showed roughly equal performance for predicting TGF$\beta$ treatment in the appropriate samples. Overall, analyses of these nine datasets indicate that BD-Func can provide similar quality results as GSEA and IPA.

## BD-Func can determine the accuracy of novel predictive models

Six breast cancer datasets were also used to test the robustness of the progesterone receptor (PGR) signature (*Anders et al., 2008*; *Chin et al., 2006*; *Finak et al., 2008*; *Huang et al., 2003*; *The Cancer Genome Atlas Network, 2012*). Unfortunately, neither BD-Func, GSEA, nor IPA could predict progesterone status in all seven patient populations (Table 3). To be fair, the original *Claus et al. (2008)* dataset was used to define progesterone receptor status in meningioma samples whereas the novel datasets tested were all breast cancer samples (where testing for over-expression of progesterone receptor is common (*Bardou et al., 2003*)). Nevertheless, BD-Func is designed to be able to utilize custom gene signatures with activated and inhibited, so we defined a novel progesterone receptor signature using the expO dataset (GEO Series GSE2109). This signature can identify progesterone

**Table 1 Recovery of known perturbations for selected MSigDB oncogenic genes.** BD-Func "Fold-Change" corresponds to analysis of fold-change values calculated between the perturbed and unperturbed groups. BD-Func "Intensity" corresponds to calculation of activity vs. inhibition score for each sample in the dataset followed by a comparison in the distribution of test statistics for all of the samples.

| | BD-Func (Fold-change) | BD-Func (Intensity) | GSEA | IPA |
|---|---|---|---|---|
| TGFβ (E-TABM-420) | Yes | Yes | Yes (UP) Yes (DN) | Yes |
| mTOR – N4 (GSE5824) | Yes | Yes | Yes (UP) Yes (DN) | No |
| P53 – V2[a] (GSE1676 ) | Yes | Yes | Yes (UP) No (DN) | Yes |
| BRCA1 (GSE4754) | Yes | Yes | Yes (UP) Yes (DN) | No |
| PGR (GSE9438) | Yes | Yes | No (UP) No (DN) | No |

Notes.
[a] [1]p53 signal changes with sign matching p53 expression (in this case P53_DN indicates genes down-regulated by knock-down of p53, not genes negatively related by p53).

**Table 2 Prediction of TGFβ activity in novel datasets.**

| | BD-Func (Fold-change) | BD-Func (Intensity) | GSEA | IPA |
|---|---|---|---|---|
| GSE1724 | Yes | Yes | Yes (UP) No (DN) | Yes |
| GSE1805 | No | No[a] | No (UP) Yes (DN) | No |
| GSE6653 | Yes | Yes | Yes (UP) Yes (DN) | Yes |
| GSE17708 | Yes | Yes | Yes (UP) No (DN) | Yes |

Notes.
[a] Qualitatively detected at 2 h but not 4 h, but activity is not significant with $p$-value $< 0.05$ for either time-point.

receptor positive and negative patients for all 7 cohorts (1 meningioma and 6 breast cancer datasets), so it is robust enough for application to multiple cancer types.

Another unique feature of BD-Func is the ability to use the activation versus inhibition test statistic as a classifier to define a predictive model. If a $t$-test statistic of 2 is used for the cut-off of distinguishing the positive and negative populations (roughly corresponding to a $p$-value $< 0.05$), it is clear that the MSigDB signature is extremely accurate at predicting PGR status in the original dataset but not in the breast cancer datasets (Table S1). Likewise, the TGFβ signature could differentiate between the treated and untreated groups if the test statistic of 2 was used as the threshold to distinguish the groups (Fig. S2). However, this threshold does not work well in all circumstances: unlike the analysis of fold-change values, the $p$-value (for any statistical method) is not always the ideal statistic for assessment of functional enrichment on intensity values. For example, the mTOR and BRCA1 signatures

**Table 3 Prediction of progesterone receptor status in patient samples.** In this table, BD-Func is used to analyze fold-change values between PR+ and PR− patients.

| Cohort | BD-Func (MSigDB) | BD-Func (COH) | GSEA (MSigDB) | IPA |
|---|---|---|---|---|
| GSE9438 (N = 31) | Yes | Yes | No (UP) No (DN) | No |
| (Huang et al., 2003) (N = 88) | No | Yes | No (UP) No (DN) | No |
| (Chin et al., 2006) (N = 117) | No | Yes | No (UP) No (DN) | No |
| (Anders et al., 2008) (N = 73) | No | Yes | No (UP) No (DN) | No |
| (Finak et al., 2008) (N = 53) | No | Yes | No (UP) No (DN) | No |
| expO (N = 256) | No | Yes | No (UP) No (DN) | No |
| TCGA (N = 739) | No | Yes | No (UP) No (DN) | No |

**Notes.**

MSigDB = CLAUS_PGR_POSITIVE_MENINGIOMA signatures. COH = novel PGR signature developed in this study.

(Fig. 2B) show appropriate changes in test statistics that clearly distinguish treated and untreated groups, but activation and inhibition can't be defined based upon a pre-defined cut-off for the test statistic value (e.g., 2 or −2). For this reason, we provide an ANOVA $p$-value to quantify the difference in test-statistic between groups, where the test statistic serves as a score for a second calculation of statistical significance.

Additionally, we believe that predictive statistics are a useful method for accessing BD-Func scores for individual samples within large patient populations. In order to quantify the accuracy of the model without depending on a pre-defined cut-off, BD-Func produces receiver operating characteristic (ROC) plots for each cohort and the area under the curve (AUC) is calculated for each ROC plot (where a 100% accurate model would have an AUC = 1.00) (Fig. S3, Table S1). The superior performance of the novel PGR signature on the breast cancer cohorts becomes even clearer when these predictive statistics are compared for the two models (Fig. 3, Fig. S4, and Table S2). Importantly, the novel signature showed the same level of accuracy on the TCGA breast cancer dataset as the original expO dataset. This is significant because the TCGA dataset is over twice as large as the expO dataset, and the TCGA dataset utilizes RNA-Seq while the expO dataset utilizes microarrays to quantify gene expression. In other words, this shows that BD-Func is capable of producing very robust predictions that translate across different genomic technologies.

### BD-Func accurately quantifies LBH589 activity using a novel signature

LBH589 (panobinostat) is a histone deacetylase inhibitor that has been previously shown to suppress the proliferation of aromatase inhibitor resistant breast cancer cells, which was

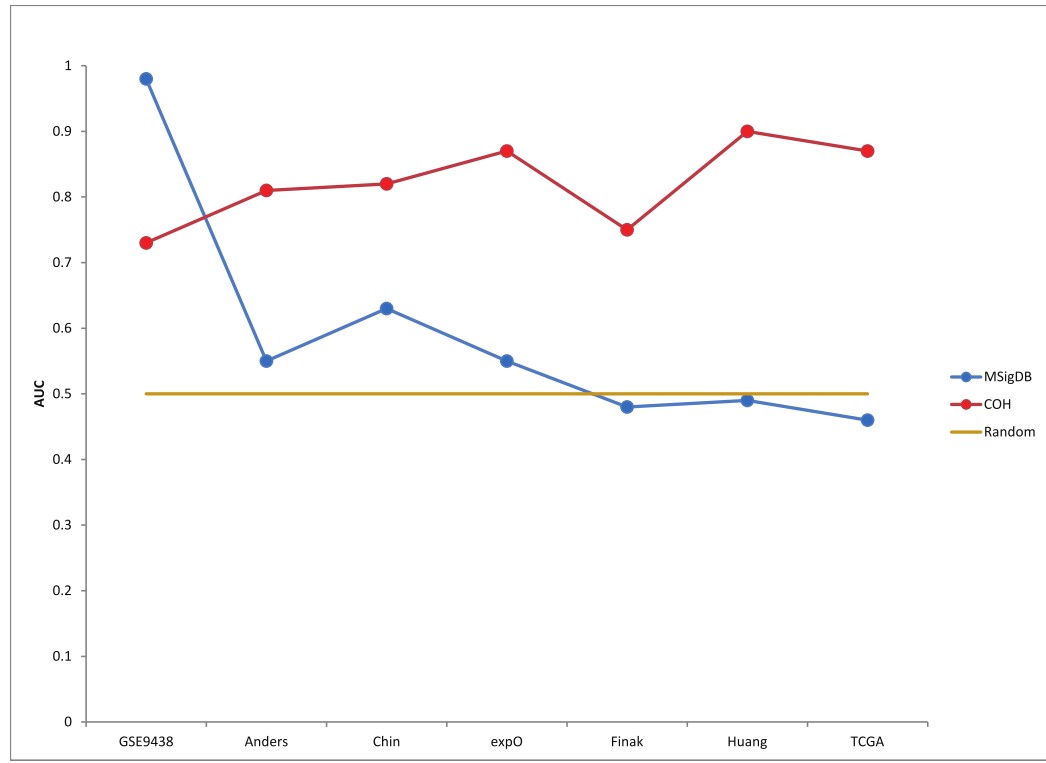

**Figure 3 Custom progesterone signature outperforms MSigDB signature on breast cancer patients.** AUC (Area Under the Curve) values from the ROC plots for each patient cohort are displayed. Although the MSigBD gene set shows extremely high accuracy for the original meningioma dataset, it shows essentially random predictive power for the 6 larger breast cancer datasets. On the other hand, a custom progesterone receptor signature defined using the expO dataset shows high accuracy for all 7 cohorts, and the high accuracy is maintained even in the largest cohort (TCGA).

a conclusion supported in part by functional enrichment analysis of commonly affected genes from 3 cell line experiments (*Kubo et al., 2013*). Gene lists derived from these cell line experiments can be easily used to define a BD-Func signature, so we hypothesized that the results from this previous *in vitro* cell culture study could be used to predict LBH589 activity *in vivo* in an animal study. Specifically, we asked if the signature defined based upon LBH589 treatment in 3 cell lines (H295R, MCF-7her2, HeLa) could detect LBH589 activity in a mouse xenograft from a different cell line (MCF-7aro xenograft treated with EXE). Indeed, BD-Func correctly used the cell line LBH589 signature to identify common gene expression changes in the tumours treated with LBH589 and EXE compared to the mice that were only exposed to EXE treatment (Fig. 4).

## DISCUSSION

Comparison of BD-Func to GSEA indicated that BD-Func can provide similar oncogenic signature predictions with much shorter run time (Table S3) and a more direct comparison of genes that are expected to be up- or down-regulated by the oncogenic regulators. One limitation to BD-Func is that it can only conduct functional enrichment for regulators with genes that are **both** up- and down-regulated, so there are many gene lists in MSigDB that
**A.**

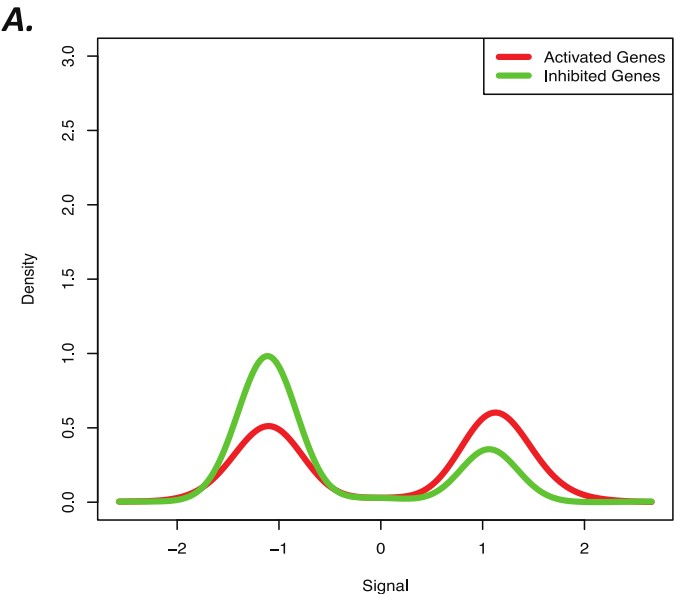

**B.**

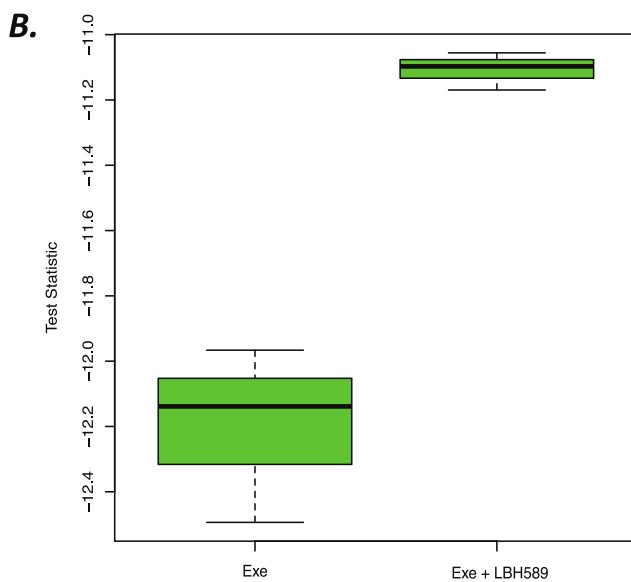

**Figure 4 Novel cell line LBH589 signature can accurately detect drug activity *in vivo*.** (A) BD-Func density plot for fold-change values for activated and inhibited genes for Exe + LBH589 vs. Exe alone tumors. At a population level, the Exe + LBH589 tumors show higher expression of activated genes whereas the Exe tumors show increased expression of inhibited genes ($p = 2.0 \times 10^{-15}$). (B) BD-Func box-plot for single-sample signature scores. EXE alone shows the greatest inhibition of LBH-related gene expression whereas EXE + LBH shows less inhibition of LBH-related gene expression.

cannot be analyzed using BD-Func (which instead should be analyzed using a tool such as GSEA). It is also worth noting that BD-Func can work with a wide range of sizes of gene lists for activated and inhibited genes (Table S4), but we would recommend using at least a few dozen genes when defining custom signatures.

Comparison of the BD-Func oncogenic signatures to the IPA upstream regulators also showed that both programs could provide similar performance, which is not surprising giving the design of that module in IPA. One benefit to utilizing IPA is that IPA has a curated database which lists with a wider variety of regulators than the MSigDB oncogenic signatures that can be analyzed in BD-Func. In contrast, one major benefit to using BD-Func is the greater theoretical range of applications. For example, BD-Func provides an enrichment file for Gene Ontology (GO) categories (*Ashburner et al., 2000*), but IPA does utilize this same strategy of analyzing functional ontologies by comparing the expression of positively and negatively regulated genes.

The Connectivity Map is a commonly used tool to study gene expression profiles for drugs and other chemical perturbations (*Lamb et al., 2006*). There are no LBH589/panobinostat treatments in the Connectivity Map database (although this database can certainly provide other useful information), so BD-Func provides a unique opportunity to test for gene signatures that show a strong positive or negative correlation with novel drug treatments (such as LBH589). Additionally, BD-Func is compatible with any gene mapping (in this case, gene symbol), whereas the Connectivity Map requires users to define their signatures in terms of HG-U133A probes. For example, affected gene

symbols had to be converted to HG-U133A probes for this analysis. We believe that being able to define signatures based upon gene symbol is a substantial practical benefit.

BD-Func also calculates a test statistic to represent functional activation or inhibition for each individual sample in a dataset, and this study shows how this statistic can be directly used as a classifier that can be used to quantify the predictive power of a given functional model. More specifically, this study shows the utility of using BD-Func for applying two novel predictive models (for progesterone receptor status in patients and for LBH589 drug treatment). The LBH589 signature provided biological confirmation that the results from an *in vitro* model can indeed apply to validation experiments *in vivo*. This is important because our hope is that the streamlined analysis, simple input file design, and BD-Func discussion board can be used to help scientists quickly and easily share novel predictive models. In short, BD-Func provides a novel framework for functional enrichment (by comparing the relative expression of activated versus inhibited genes) that is freely available with a user interface that is accessible to biologists without any coding experience. The results of this paper show that BD-Func provides accurate predictions matched by other popular tools, which make it a useful complement to standard analysis using tools like GSEA or IPA.

## ACKNOWLEDGEMENTS

We would like to thank Christine Brown, Mike Barish, and Thanh Dellinger for discussions that led to the creation of this algorithm. We would like to thank Xiwei Wu, Zheng Liu, and two anonymous reviewers for discussions regarding the BD-Func algorithm. We would like to thank the City of Hope Functional Genomics Core for processing the microarray data.

### Funding

This work was supported by grants from the National Institutes of Health [Comprehensive Cancer Center Grant P30 CA33572], Susan G. Komen for the Cure [KG080161 to SC] and City of Hope National Medical Center institutional funding. The funders had no role in study design, data collection and analysis, decision to publish, or preparation of the manuscript.

### Grant Disclosures

The following grant information was disclosed by the authors:
NIH: P30 CA33572.
Susan G. Komen for the Cure: KG080161.
City of Hope National Medical Center.

### Competing Interests

The authors declare there are no competing interests.

## Author Contributions

- Charles D. Warden conceived and designed the experiments, performed the experiments, analyzed the data, wrote the paper, designed the algorithm.
- Noriko Kanaya conceived and designed the experiments, performed the experiments, contributed reagents/materials/analysis tools, wrote the paper.
- Shiuan Chen conceived and designed the experiments, contributed reagents/materials/analysis tools, wrote the paper.
- Yate-Ching Yuan conceived and designed the experiments, wrote the paper.

## Animal Ethics

The following information was supplied relating to ethical approvals (i.e., approving body and any reference numbers):

All animal research procedures were approved by the City of Hope Institutional Animal Care and Use Committee.

## Microarray Data Deposition

The following information was supplied regarding the deposition of microarray data:

GSE47346.

## Supplemental Information

Supplemental information for this article can be found online at http://dx.doi.org/10.7717/peerj.159.

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
