# Peer review of "BD-Func: a streamlined algorithm for predicting activation and inhibition of pathways"

_PeerJ, doi:10.7717/peerj.159_

## Round 0.1 · original submission · Minor Revisions

I apologize for the unusually lengthy review process. This was in part due to the time of year and difficulty of finding reviewers who could review within the required time frame.

Please consider the reviewers suggestions carefully to ensure maximum readability of the manuscript. In particular please pay attention to the sections asking for additional detail on selection of approaches used.

Reviewer 1 ·

Basic reporting

"No Comments"

Experimental design

"No Comments"

Validity of the findings

"No Comments"

Additional comments

Reviewers' comments:
Reviewer:
Major comments:
The manuscript "BD-Func: A Streamlined Algorithm for Predicting Activation and Inhibition of Pathways" presents BD-Func (BiDirectional FUNCtional Enrichment) a new algorithm for the functional enrichment of microarray and RNAseq data based on lists of genes that are known to be activated versus inhibited in a pathway or by a regulatory molecule. The authors effectively compare the accuracy of BD-Func with that of GSEA and IPA (“Upstream Regulator” module); two very popular enrichment analysis tools. For the comparison the authors not only selected several MSigDB oncogenic signatures but also novel datasets. The results obtained highlight which are the strengths and limitations of using BD-Func tool however some comments should be addressed.
1. Since there are lots of functional enrichment tools is not obvious in the
introduction why the authors selected GSEA and IPA tools to compare them
with BD-Func tool.
2. Discussion, 1st paragraph- “One limitation to BD-Func is that it can only
conduct functional enrichment for regulators with genes that are both up- and
down-regulated, so there are many gene lists in MSigDB that cannot be
analyzed using BD-Func”. The authors should explain this limitation since as
they referred that many gene lists will be left out of the analysis. Which other
tools as a suggestion of the authors can be used in this particular case.

Minor comments:
Introduction:
1st paragraph- “Systems-level analysis of the combined expression pattern of multiple genes can sometimes be more informative than the expression pattern of an individual gene, and there are a number of tools to calculate functional enrichment of differentially expressed genes.”
This sentence should be revised. The information obtained by the combined expression pattern of multiple genes or their products is always more informative than the expression pattern of an individual gene (reductionist approach) unless, e.g., previous data situate this gene as a biomarker in the context of the study.
1st paragraph- “However, many functional enrichment tools would expect all the members of the pathway to behave similarly.”
Is important to mention some of these tools, at least the most used, to guide the reader and show him/her which tools do not perform a similar analysis to BD-Func.
2nd paragraph- “Most functional enrichment tools either require upstream filtering of gene lists or a comparison of signal intensities between two groups.”
Same as before which tools require upstream filtering of gene lists and which require a comparison of signal intensities.
Results:
BD-Func Shows Equal or Greater Performance to GSEA and IPA for Functional Enrichment:
1st paragraph- “Nevertheless, the significance of this analysis is that BD-Func can detect accurately detect perturbation of all of these biomarkers on datasets where we know that these specific genes will be altered”.
Change sentence to: Nevertheless, the significance of this analysis is that BD-Func can accurately detect perturbation of all of these biomarkers on datasets where we know that these specific genes will be altered.

Reviewer 2 ·

Basic reporting

In "BD-Func: A Streamlined Algorithm for Predicting Activation and Inhibition of Pathways", authors present a fast method (employing t-test, and ANOVA) which, using gene expression data and GO annotations, can predict the functional enrichment of input activated and inhibited genes. Software for the method is also made available online.

Although the written English is of good enough standard, but article seems to be written in hurry. In my opinion, the background to the need for this methodology is not appropriately described while there is plenty of vague, un-necessary and out-of-context text scattered around.
Some specific comments:
1 - In the abstract, the sentence "… functional predictions on a single sample (and compare scores for individual samples across multiple groups), and BD-Func can provide predictive statistics and receiver operating characteristic (ROC) …”
should be written as “… functional predictions on a single sample, compare scores for individual samples across multiple groups, and provide predictive statistics and receiver operating characteristic (ROC) …”
2 – First sentence on page 2 is vague and does not make strong connection with functional enrichment methods referenced there in. Importantly, there are 25+ citations dumped almost at the start of the paper, which does not seem reasonable. In my opinion, these should be reduced significantly, citing only the most relevant ones and perhaps citing some appropriate review article of such methods, if available.
3- Again in page 2, the sentence “… in a single sample is a unique feature only present in a limited number of functional genomic tools …” How does this stand with the relevant statement made in the abstract (given in (1))?
4- Page 6: “For the Connectivity Map analysis…. ” Seems like out of place or at least without much connection established with the ongoing section.

Experimental design

Author use t-statistics and ANOVA output to compare groups of activated and inhibited genes, for enrichment analysis. It might well be, as authors claim, an efficient way of performing enrichment analysis for the case of activated and inhibited genes and it might be useful for the community. My concern is that the research question answered by this methodlogy was poorly defined, and methodology is not well explained at all. It is hard to find significant novelty in the method, at least from the way it is explained in the manuscript. Authors must explain the new method in much more detail and step-wise fashion so that reader can get the sense of it and can appreciate the novelty and application. A lot has been left for the reader to explore. Introduction and Methods sections need to be re-written with much clarity and detail

Validity of the findings

Authors compare their approach to two well-known methods, i.e., GSEA and IPA, using signatures from MSigDB and other novel datasets. Overall comparison is good on these limited data sets, but I am unable to make an authentic statement about the wider applicability of the approach. Also, in comparison to existing methods, BD-Func seems to have limited application, i.e., only the cases related to both up and down regulated genes.

Additional comments

The manuscript needs major revisions in order to be considered for publication. Besides the overall tidying-up of the manuscript, by removing un-necessary text and citations, Introduction and Methods section need major re-write.

---

## Round 0.2 · accepted · Accept

Thank you for your revisions and patience during the unusually long review period due to summer vacations.